# Do Carbon Nanotubes and Asbestos Fibers Exhibit Common Toxicity Mechanisms?

**DOI:** 10.3390/nano12101708

**Published:** 2022-05-17

**Authors:** Suchi Smita Gupta, Krishna P. Singh, Shailendra Gupta, Maria Dusinska, Qamar Rahman

**Affiliations:** 1Department of Systems Biology and Bioinformatics, University of Rostock, 18051 Rostock, Germany; suchi.smita@uni-rostock.de (S.S.G.); krishna.singh@uni-rostock.de (K.P.S.); shailendra.gupta@uni-rostock.de (S.G.); 2Health Effects Laboratory, Department of Environmental Chemistry, NILU-Norwegian Institute for Air Research, 2007 Kjeller, Norway; mdu@nilu.no; 3Amity Institute of Biotechnology, Amity University, Lucknow 226028, India

**Keywords:** carbon nanotubes, asbestos, exposure, fiber toxicity, toxicity pathways

## Abstract

During the last two decades several nanoscale materials were engineered for industrial and medical applications. Among them carbon nanotubes (CNTs) are the most exploited nanomaterials with global production of around 1000 tons/year. Besides several commercial benefits of CNTs, the fiber-like structures and their bio-persistency in lung tissues raise serious concerns about the possible adverse human health effects resembling those of asbestos fibers. In this review, we present a comparative analysis between CNTs and asbestos fibers using the following four parameters: (1) fibrous needle-like shape, (2) bio-persistent nature, (3) high surface to volume ratio and (4) capacity to adsorb toxicants/pollutants on the surface. We also compare mechanisms underlying the toxicity caused by certain diameters and lengths of CNTs and asbestos fibers using downstream pathways associated with altered gene expression data from both asbestos and CNT exposure. Our results suggest that indeed certain types of CNTs are emulating asbestos fiber as far as associated toxicity is concerned.

## 1. Introduction

The last few decades have seen an explosion of thousands of engineered nanomaterials synthesized with precise size, shape and structural specifications. These nano-structured materials are composed of different base materials (mainly carbon, silicon and metals, such as gold, silver, titanium, cadmium and selenium) and have numerous novel and useful properties as they have substantially more reactive atoms on their surfaces compared to similar materials in the micro size range. Among the newly designed materials, carbon nanomaterials are the most used engineered nanomaterials in the form of nanoparticles, nanowires or nanotubes. Carbon nanotubes (CNTs) are hollow nanofibers of single (single wall carbon nanotube—SWCNT) or multiple (multi-wall carbon nanotube—MWCNT) layers of carbon atoms arranged in a honeycomb-like structure with two dimensions sized in nanoscale, i.e., 1–100 nm while the third dimension is very long (sometimes up to several millimeters in lengths), comparable to fibrous materials [1]. Due to their unique electrochemical properties, effectiveness in heat conductivity, unusual strength (10-fold stronger than steel and 1.2-fold harder than diamond) and very light weight [2], CNTs have emerged as highly exploitable materials for a wide spectrum of industrial and medical applications [3,4,5,6,7]. The global market of CNT, which was around $50.9 million in 2006 [8] had increased to $4.47 billion by 2018 and is expected to reach around $15 billion by 2026 [9]. The properties responsible for the exponential growth in the application and production of CNTs also raise potential concerns about potential adverse health effects. In particular, the fiber-like structure, high aspect ratios (high length: width ratio), physiochemical durability and presumed bio-persistency in lung tissues are linked to past experience with hazardous asbestos fibers, have brought these materials under scrutiny [10,11,12,13,14,15,16].

Various epidemiological and animal studies have shown that other non-asbestos fibers, e.g., erionite, fluoro-edenite, organic fibers from plant origin and manmade vitreous fibers that are more than 5µm in length and narrow enough to reach the distal lung upon inhalation might conform to the ‘fibre pathogenicity paradigm’ and might be associated with development of malignant mesothelioma [17,18,19,20,21,22,23,24,25]. These studies indicate that fiber morphology is one of the main decisive factors responsible for malignant mesothelioma in the exposed population. Several groups have pointed out the potential of CNTs to induce malignant mesothelioma in a way similar to asbestos fibers [11,24,26,27,28,29,30,31,32].

Earlier studies revealed a high level of genetic damage in the lymphocytes of workers exposed in an asbestos factory, particularly among workers who also smoked [33]. In one study on male Fischer rats, Sakamoto and colleagues assessed the carcinogenic hazard of MWCNTs (1 mg/kg body weight) compared to crocidolite (blue asbestos) (2 mg/kg body weight) exposure. They found that after 37–40 weeks, 6 of the 7 MWCNT-treated animals (85.7%) died or became moribund due to intraperitoneally disseminated mesothelioma associated with bloody ascites, while all crocidolite-treated rats survived for 52 weeks without any changes except deposition of asbestos. Their results suggest that MWCNTs are capable of inducing mesothelioma at a high rate in normal male rats compared to asbestos [24]. Similarly, Takagi and colleagues have demonstrated that MWCNTs form fibrous or rod-shaped particles of length around 10–20 µm and induce mesothelioma in a similar way to crocidolite [34].

Generally, the harmful effects of CNTs arise from the combination of various parameters that are known to be associated with fiber pathogenicity, and the following four are of great concern: (a) high surface to volume ratio, (b) fibrous needle-like shape that resembles asbestos, (c) bio-persistent nature of nanotubes and (d) capacity to adsorb toxicants/pollutants on the surface. The first two parameters are invariably the same for any type of fibrous structures capable of inducing acute pleural inflammation while the last two depend on the chemical nature of fibers. The pathogenicity paradigm of long, thin and bio-persistent fibers is shown in Figure 1.

CNTs have been listed by the International Chemical Secretariat (ChemSec) as so-called SIN (‘Substitute It Now’) chemicals to be restricted or banned in the EU in November 2019 [35], www.sinlist.chemsec.org accessed on 15 January 2022). However, there are numerous types of CNTs and they differ substantially in physico-chemical properties, and so cannot be evaluated only on the basis of chemical composition [36]. MWCNT-7 has been already classified as possibly carcinogenic to humans by IARC (Group 2B) [37]. CNTs have a high aspect ratio which resembles that of asbestos and other fibers causing lung cancer and mesothelioma.

High aspect ratio nanomaterials (HARN) are defined as nanofibers with two similar external dimensions and a significantly larger third dimension (aspect ratio of 3:1 or greater) and substantially parallel sides’ [38]. Not all HARNs are associated with mesothelioma [39], and aspect ratio is not the only factor responsible for potential pathogenicity.

## 2. Do Asbestos and CNTs Have the Same Mechanism of Pathogenicity?

To identify the commonality between asbestos and CNT pathogenicity, it is important to understand the mechanism by which asbestos induces asbestosis, bronchogenic carcinoma and mesothelioma in humans. Bernstein and colleagues proposed that the mechanisms of lung disease caused by certain fibers are numerous and include mainly oxidative stress, inflammation and direct or indirect genotoxicity [40]. As reviewed by Nagai and Toyokuni, there are four main hypotheses (highlighted in Figure 2), regarding the mechanisms of asbestos-induced pathogenicity [41]. (a) oxidative stress, (b) chromosome tangling, (c) adsorption and (d) chronic inflammation. Various studies were conducted in the past to support these hypotheses. Interestingly, several studies on CNTs have reported similar mechanisms of action as discussed in the following text.

### 2.1. Oxidative Stress Theory

The first theory postulates the generation of reactive oxygen and nitrogen species (ROS/RNS) as a consequence of injury to pleural mesothelioma cells injury due to exposure to asbestos fibers [42,43,44,45,46,47]. Several studies also showed the formation of free radicals, accumulation of peroxidative products and depletion of cell antioxidants in the keratinocytes and bronchial epithelial cells exposed in vitro to SWCNTs [48,49,50] and MWCNTs [51,52,53,54,55]. Like asbestos, SWCNTs/MWCNTs contain high levels of Fe, Ni, Co, Mo and other transition metal impurities which are known to induce ROS/RNS formation. These metals or metal mixtures are common components used in CNT synthesis, and all of these have demonstrated toxicity [56,57]. In general, each CNT sample invariably contains three classes of residual impurities from the synthesis process: metals, organics, and growth support material. It is worth mentioning that even purified grade CNTs still contain 1–5% residual metal by mass [58]. The residual organics include various forms of bulk carbon (amorphous soot particles or micro-structured graphite sheets) and other residual organic molecules. Aluminate and silicate residues are shown to be present on CNTs as they are used as materials to support the catalyst or growth region [59]. Furthermore, functionalization of CNTs by the addition of certain surface molecule groups can modify their specific toxicity [60,61]. Mainly the metals, organics and growth support materials present on CNTs, when they come into the contact with cells, result in oxidative stress [57,62,63,64] and damage cellular macromolecules.

Oxidative stress induced by asbestos activates several signaling cascades that are necessary for cell proliferation, such as, MAPK, NF-ĸB, AP-1 and ERK 1/2 [65,66,67,68,69] in dose- and time-dependent manner. AP-1 and NF-ĸB (redox-sensitive transcription factors), which are activated by asbestos/CNT exposure, regulate expression of several genes involved in inflammation, proliferation, apoptosis and the carcinogenesis process. Several studies report the mitochondria-mediated production of ROS, their localization and the resulting damage in response to asbestos/CNT exposures [70,71,72,73,74,75,76,77,78].

In their study on male CD-ICR mice, Yang and colleagues showed accumulation of SWCNTs in the liver, spleen and lung 90 days after a single tail vein injection with 40 µg, 200 µg or 1.0 mg of SWCNTs per mouse (10–30 nm diameter and 2–3 µm length, containing impurities (wt%): Fe 0.4, Ni 3.0 and Y 1.3) [79]. Although no abnormal symptoms were observed, reduced glutathione (GSH) levels were found in the liver and lungs of all exposed groups along with increases in malondialdehyde (MDA) levels in the liver and lung indicating that SWCNTs induce oxidative damage. In another study, Murray and colleagues measured the dermal toxicity of purified/unpurified SWCNT both in vitro and in vivo using EpiDerm FT engineered skin, murine epidermal cells (JB6 P+) and immune-competent nude SKH-1 mice. Upon SWCNT exposure, the EpiDerm FT engineered skin, showed enhanced epidermal thickness due to accumulation and activation of dermal fibroblasts. The unpurified SWCNTs (with 30% Fe) exposure to JB6 P+ cells resulted in an increase in hydroxyl radical concentration. Although no significant changes were observed in the AP-1 activation with the partially purified SWCNTs (with 0.23% Fe), NF-ĸB was activated in a dose-dependent manner by exposure to both unpurified and partially purified SWCNTs. Reduction in glutathione concentration and oxidation of protein thiols/carbonyls were observed in SKH-1 mice due to oxidative stress when they were exposed to unpurified SWCNTs (5 days, with daily doses of 40 µg/mouse, 80 µg/mouse or 160 µg/mouse) [80]. These data highlight the role of SWCNT-mediated oxidative stress in causing dermal toxicity.

In another study with industrial MWCNTs (6–24 nm in diameter and 2–5 µm in length, containing 0.4 wt% Fe impurity), Thurnherr and colleagues exposed A549 human lung epithelial cells and T lymphocytes for 2 h. They observed concentration-dependent levels of ROS and decreased mitochondrial activities; however, no morphological changes in the cells were noticed [81]. Fenoglio and co-workers exposed murine alveolar macrophages (MH-S) with two distinct sets of MWCNTs with similar length (<5 µm) but different diameters (9.4 and 70 nm). Both samples were internalized in the MH-S cells; however, the MWCNTs with thin diameter generated a high level of ROS as measured by DCF-DA fluorescence in comparison to those of larger diameter on a mass-dose basis, confirming that thin MWCNTs are more toxic [82].

In a study on rat epithelial cells exposed to MWCNTs, researchers found induction of mitochondrial apoptotic factors responsible for the reduced cellular ATP contents due to the collapse of mitochondrial membrane integrity [83]. Zhou and co-workers have reported changes in the mitochondrial transmembrane potential caused by localization of PL-PEG functionalized SWCNTs in mitochondria of both tumor as well as normal cells [73], ultimately leading to exaggerated ROS production due to collapse of mitochondrial membrane potential. A dose- and time-dependent decrease in the mitochondrial membrane potential due to generation of intracellular oxygen species was observed in rat macrophages and A549 cells exposed to commercially available SWCNTs and MWCNTs with metal impurities whereas no effect was detected with the CNTs treated with acid to remove residual transitional metal traces [84]. These findings suggest that cells exposed to CNTs will have dysfunctional mitochondrial activities. However, a study in A549 cells, in which the ROS formation induced by MWCNTs was independent of mitochondrial activity as measured by MTT assay, suggests that there are also other factors that generate ROS after CNT exposure [85]. The ROS generated on exposure to fibers were shown to cause lipid peroxidation as indicated by the synthesis of mutagenic compounds such as malondialdehyde (MDA) and 4-hydroxynenal. MDA formation was observed after exposure of HUVEC cells [86] and A549 cells [85] to MWCNTs. High levels of MDA were also observed in rat blood and bronchoalveolar lavage fluid after CNT exposure through intraperitoneal/intravenous injections or via intratracheal instillation [86,87,88].

Several studies have shown that exposure to fibers causes depletion of intracellular antioxidant defense by the generation of free radicals. The A549 cells when exposed with crocidolite-like silicate fibers inhibited the pentose phosphate pathway (a key antioxidant intracellular system) by the inhibition of glucose-6-phosphate dehydrogenase [89]. The same cell lines when exposed to MWCNTs were also found to have reduced catalase and glutathione activity [85]. SWCNTs also elicit the same effect in rat lung epithelial cells [90].

All these studies clearly indicate that both asbestos and CNT fibers share common mechanisms of oxidative stress upon exposure.

### 2.2. Chromosome Tangling

Several studies have demonstrated the disruption of chromosomal structure, due to asbestos fiber exposure at the time of mitosis, resulting in the inheritance of abnormal chromosome by the daughter cells including mesothelial cells [91,92,93,94,95,96]**.** Trisomy of chromosome 11 was found in six out of eight Syrian hamster embryo cell lines derived immediately after asbestos exposure [97]. Jiang and colleagues have demonstrated the direct interaction of asbestos fibers with the chromosomes [98]. Exposure of cultured cells with chrysotile have been shown to cause double strand breaks in DNA [99,100] along with intrachromosomal deletion and DNA mutations [101]. Cortez and co-workers have identified aneuploid cell formation, increased number of cells in G2/M phase and cells with multipolar mitosis in an in vitro study of chrysotile-exposed lung cancer cells [102]. Various types of chromosomal damage may be observed in cells presented with asbestos fibers, including chromosomal breaks and fragments (micronuclei), exchange of chromosomal, lagging chromosomes, segments between two chromosomes and chromosomal mis-segregation [103]. Some of the asbestos fibers genotoxicity studies are highlighted in Table 1.

Similar to the asbestos fibers, several carbon-based nanomaterials exhibit genotoxicity [52,108,109]. MWCNTs were found to be genotoxic to human lung cells in vitro at occupationally relevant dose [110,111,112]. Sasaki and colleagues analyzed MWCNTs with different shape and size and found that straight MWCNTs induce more polyploidy followed by curved and tangled fibers in Chinese hamster lung cell line (CHL/IU) [112]. Siegrist and colleagues showed cell cycle disruption, mitotic spindle disruption and aneuploidy in human lung epithelial cell lines (BEAR-2B and SAEC) exposed to MWCNT-7, designed by the International Agency for Research on Cancer, and two physiochemically-altered MWCNTs [111]. SWCNTs/MWCNTs (~1–25 nm diameter and ~500–1000 nm in length) have been shown to disrupt chromosomal distribution during mitosis resulting in aneuploidy in the daughter cells [113,114,115,116,117]. Li et al. observed that SWCNTs preferentially bind to the major groove of DNA with GC preference [118]. The inhibition of DNA duplex association and formation of telomeric i-motif was also observed due to the binding of SWCNTs at the 5′-end major groove of DNA [119]. Mangum and co-workers showed the joining of daughter cells of alveolar macrophages by the formation of carbon bridges composed of CNTs [120]. CNT exposure significantly increased micronuclei in human primary small airway epithelial cells (SAEC) indicating aneugenic events triggered by CNTs [121]. Sargent and co-workers have shown mitotic spindle aberrations in SAEC exposed to 24, 48 and 96 µg/cm^2^ SWCNTs [114]. Moreover, fragmented centrosomes, disrupted mitotic spindles and aneuploidy were observed in the SAEC exposed to SWCNTs for 24–72 h at doses equivalent to 20 weeks of exposure at the permissible exposure limit for particulates [113]. In a study by Zhu and colleagues on mouse embryonic stem cells, MWCNTs were shown to increase the mutation frequency by 2-fold compared to the spontaneous mutation frequency. The increased expression of the key base excision repair pathway enzyme 8-oxoguanine-DNA glycosylase 1 (OGG1) and double strand break repair proteins (Rad51 and XRCC4) was also observed with MWCNT exposure [122].

These studies clearly suggest that both asbestos and CNTs induce genotoxicity due to disruption of chromosomal structure, mutations and double-strand DNA breakage. Table 2 shows some of the important genotoxicity studies after the exposure of SWNCTs/MWCNTs in both in vitro and in vivo settings.

### 2.3. Adsorption Theory

The adsorption theory postulates the surface reactivity of fibers for certain proteins and molecules. Due to the surface reactivity, many carcinogenic molecules may get adsorbed on fiber surfaces from various environmental matrices. These molecules, once released into the cell after fiber internalization, cause the pathogenicity. MacCorkle and co-workers demonstrated that internalized asbestos fibers have high affinity to bind with proteins involved in the regulation of cell cycle, cytoskeleton and mitotic process to induce aneuploidy and genotoxicity. However, pre-coated asbestos fibers with protein complexes did not induce aneuploidy without affecting fiber uptake by the cells [128]. Various known mutagens such as benzo(a)pyrene from cigarette smoke have high affinity for asbestos [129,130,131,132]. Jiang et al. also showed that chrysotile fibers accumulate iron from surrounding tissue, probably via a hemolysis process and that this catalytic iron plays an important role in asbestos-induced carcinogenesis [98].

Like asbestos, toxicity of CNTs not only comes from their own structure but also from the various toxic substances adsorbed on their surfaces. Highly hydrophobic surfaces of CNTs have already been reported as strong adsorbents for various organic compounds such as polycyclic aromatic hydrocarbons [133,134,135,136,137], phenolic compounds [138,139,140], chlorobenzenes [141,142,143,144], dioxin [145,146] and other natural organic materials [147,148,149,150]. Moreover, several studies highlight the adsorption potential of CNTs for heavy metals [151,152]. Although many of these absorption studies underline a potential role for CNTs in cleaning polluted water and other environmental matrixes, many of the adsorbed compounds have also been responsible for inducing carcinogenesis and thus, the fate of these compounds on the CNTs is still a matter of concern, as it is for asbestos fibers.

### 2.4. Chronic Inflammation

The last established theory of asbestos-induced carcinogenesis suggests the role of persistent macrophage activation resulting in chronic inflammation as one of the major events associated with the disease progression. Several studies have drawn similarities between asbestos and CNTs for inducing inflammatory reactions in human lung epithelial cells [10,153,154,155,156,157]. Rydman and co-workers explored the variation between two different CNTs and asbestos in inducing pro-inflammatory reactions in C57BL/6 mice subsequent to single pharyngeal aspiration exposure [158]. In their experiment, they used long tangled, and long rod-like CNT as well as crocidolite asbestos at a dose of 10 or 40 µg/mouse and mice were sacrificed 4 and 16 hours or 7, 14 and 28 days after the exposure. The study clearly indicates that long rod-like CNT is considerably more potent to induce lung inflammation than asbestos and long tangled fibers, with the involvement of IL-1β in mediating the inflammatory processes [158].

The long fibrous foreign materials are generally captured by macrophages and entrapped within lysosomes. However, these materials are not fully phagocytosed which ultimately results in frustrated macrophages and further inducing of chronic inflammation. In many clinical studies, these immune responses with chronic inflammatory conditions were directly linked with the progression of malignant diseases. Glass fibers of length ~17 µm were found to play a major role in incomplete phagocytosis and to induce the production of the pro-inflammatory mediators NF-κB and tumor necrosis factor alpha (TNF-α) in mouse macrophages, However, short (~7 µm) fibers showed both complete phagocytosis and less expression of inflammatory mediators [159]. Hsieh and co-worker showed airway hyperactivity and air flow obstruction due to granulomatous changes in the lung parenchyma up to six months after a single instillation of SWCNTs to the intratracheal region of mice. They also identified up-regulation of cathepsin K, MMP12, chemokines C-C motif ligands (CCL2 and CCL3) and macrophage receptors (Toll-like receptor 2, macrophage scavenger receptor 1) [160]. Several previous studies also demonstrated the activation of NF-kB and AP-1transcriptional machinery that stimulates many proinflammatory cytokines, chemokines and the expression of genes involved in the inflammation and cell proliferation processes due to exposure to long fibers such as CNTs and asbestos [161,162,163,164]. Among the highly expressed genes, COX-2 is mainly associated with cell proliferation and inhibition of apoptosis [165,166,167]. Many other inflammatory genes such as TNF-α, IL-8 and IL-1β are also regulated after exposure to CNTs and other long fibers [168,169]. In a comprehensive study on the effects of pulmonary exposure to 10 commercial MWCNTs, Poulsen and co-workers found that inflammation and genotoxicity were related to dose, time and physicochemical properties [170]. More specifically, they found that MWCNTs with larger diameter and small BET surface area are associated with increased genotoxicity and inflammation.

All these studies suggest that both asbestos and certain carbon nanofibers share mechanisms underlying chronic inflammation which can mainly be attributed to their fibrous structure.

## 3. Toxicogenomics Analysis of Altered Gene Expression Due to Asbestos and CNT Exposure

Toxicogenomics is a field of science which helps in formulating hypotheses about underlying mechanisms of toxicity by merging conventional toxicology and functional genomics. Using genomics approaches, information regarding specific mechanisms at a molecular level about nanomaterial distribution or toxicity towards multiple cellular functions becomes clearer. Conventional toxicity assays are quick in predicting the impact of exposure at the phenotypic level, but fail to formulate hypotheses about how such changes affect human beings [171].

Various studies have been performed to investigate the toxic effect of asbestos and CNT’s in different organisms using toxicogenomics studies; however, the results were always controversial due to various factors such as length, diameter, surface area, purity and tendency for agglomeration and dispersion in media [172,173,174,175]. As mentioned earlier, many studies confirmed that CNTs of various size and shape induce ROS generation, immune suppression and pulmonary fibrosis that is associated with an increased risk of lung cancer. This is mainly due to the change in the gene expression profiles related to oxidative stress response, cellular transport, metabolism and cell cycle regulation in both in vivo and in vitro systems [176,177,178,179,180,181,182].

In a 104-week long carcinogenicity study, Kasai et al. found that lung carcinomas were significantly increased in a concentration- and dose-dependent manner in both male and female rats exposed to MWCNT-7 [183]. Signaling pathways in the lung induced by asbestos cause changes in gene expression, release of cytokines, blocking of mitochondrial activities and apoptosis, ultimately leading to cancer [184]. In other studies, it has been shown that when asbestos or CNTs come into the contact with macrophages, they start inducing tumor necrosis factor (TNF), an inflammatory cytokine, and cause interleukin (IL) up-regulation. The above-mentioned studies clearly show that TNF and interleukins (IL6, IL8 and IL10) are up regulated by asbestos and MWCNT treatments.

Kim and his co-workers studied toxicogenomic effects of MWCNTs and asbestos (crocidolite) on 31,647 genes at the 50% growth inhibition (GI 50) concentration on normal human bronchial epithelia (NHBE) cells [185]. These cells were exposed to asbestos and MWCNTs for 6 and 24 h. In total 1201 and 1252 genes were up-regulated, while 1977 and 1542 genes were down-regulated by both asbestos and MWCNTs, after 6 and 24 h of exposure, respectively. Interestingly, 12 mesothelioma and 22 lung cancer-related genes were differentially regulated over two-fold by both asbestos and MWCNTs exposure in comparison to the negative control, indicating the similarity between asbestos- and CNT-mediated toxicity mechanisms.

Asbestos and CNTs can also affect signaling pathway networks which regulate the expression of genes associated with inflammatory response, apoptosis and oxidative stress. Using single-cell RNAseq, Joshi and colleagues found that macrophage colony-stimulating factor receptor signaling is essential in monocyte-derived alveolar macrophages and fibroblasts during asbestos-induced fibrosis [186]. Different studies revealed that nanoparticles interact with cell membrane elements responsible for the regulation of receptor-mediated signaling pathways [187,188]. Mukherjee and colleagues proposed that SWCNTs interact with Toll-like receptor 4 (TLR4) in the absence of a protein corona through hydrophobic interactions and attract cytokine and chemokine cascades [189]. Many recent studies focus on global mRNA and ncRNA expression profiles in the blood of workers exposed to CNTs; they have found many differentially regulated genes involved in cell cycle regulation, apoptosis and proliferation similar to those affected by asbestos fibers [178,180,190].

Nymark and colleagues exposed human bronchial epithelial BEAS 2B cells to MWCNTs and asbestos and found decreased mitochondrial membrane potential (MMP) for MWCNTs at a biologically relevant dose (0.25 μg/cm^2^) and for asbestos at 2 μg/cm^2^. They also identified 330 gene signatures related to MWCNT- and asbestos-induced MMP, of which 26 were already known for mitochondrial function [190]. In another study by Poulsen and colleagues, both in vitro and in vivo experiments were performed and compared using DNA array followed by gene-specific RT-qPCR assay after the exposure of lung epithelial cells (FE1) and mouse models. In both models, oxidative stress, fibrosis and inflammation related processes were altered with different sets of associated genes [191]. In a recent study by Jiang and colleagues, effects of length, functional group and electronic structures of various types of SWCNTs were observed in a toxicogenomic assay involving A549 cells. Their toxicogenomic analyses suggest that short SWCNTs (0.5–2 μm) had a higher toxicity level than the long ones (5–30 μm) while carboxylated SWCNTs induced greater genotoxicity, chemical stress and protein damage compared with hydroxylated ones [192].

All the above studies suggest that toxicogenomic analysis is very helpful in analyzing similarities and differences between the asbestos- and CNT-associated genes and biological pathways. Together with adverse outcome pathway (AOP) analysis, describing mechanistic information on toxicology response and toxicity initiation events, toxicogenomic studies can be a useful tool for safety assessment of emerging nanomaterials [193]. We see a clear picture of CNTs following in the footprints of asbestos in terms of toxicological outcomes through similar cellular processes, cellular transport, metabolism, cell cycle regulation, stress response, immune response, inflammatory response, genotoxicity and apoptosis in both in vitro and in vivo settings [194].

## 4. Conclusions

Several possible mechanisms have been proposed to define the toxicity and carcinogenicity of fibrous materials including certain types of CNTs and asbestos fibers. These include oxidative stress, chromosomal damage, adsorption of toxicants and pollutant and chronic inflammation. These pathogenic theories include a variety of intrinsic and extrinsic factors that modulate the biological responses. Among these, we have focused on how asbestos fibers and CNTs enter non-phagocytic cells, which is important in mesothelial/epithelial cell injury. Based on current evidence, all four theories hint towards the carcinogenic outcomes and could extensively contribute to the explanation of the toxicity of asbestos and certain nanomaterials. The profiles of gene expression upon exposure to asbestos and CNTs provide important information on the pathways that are commonly shared or unique to each fibrous type. Based on this discussion, it is evident that asbestos and certain diameters and lengths of CNTs share common mechanisms for pathogenicity and thus indicate an immediate need for the designing of a protocol to regulate the industrial use of some CNTs to avoid a hazardous situation such as was witnessed with asbestos fibers.

## Figures and Tables

**Figure 1 nanomaterials-12-01708-f001:**
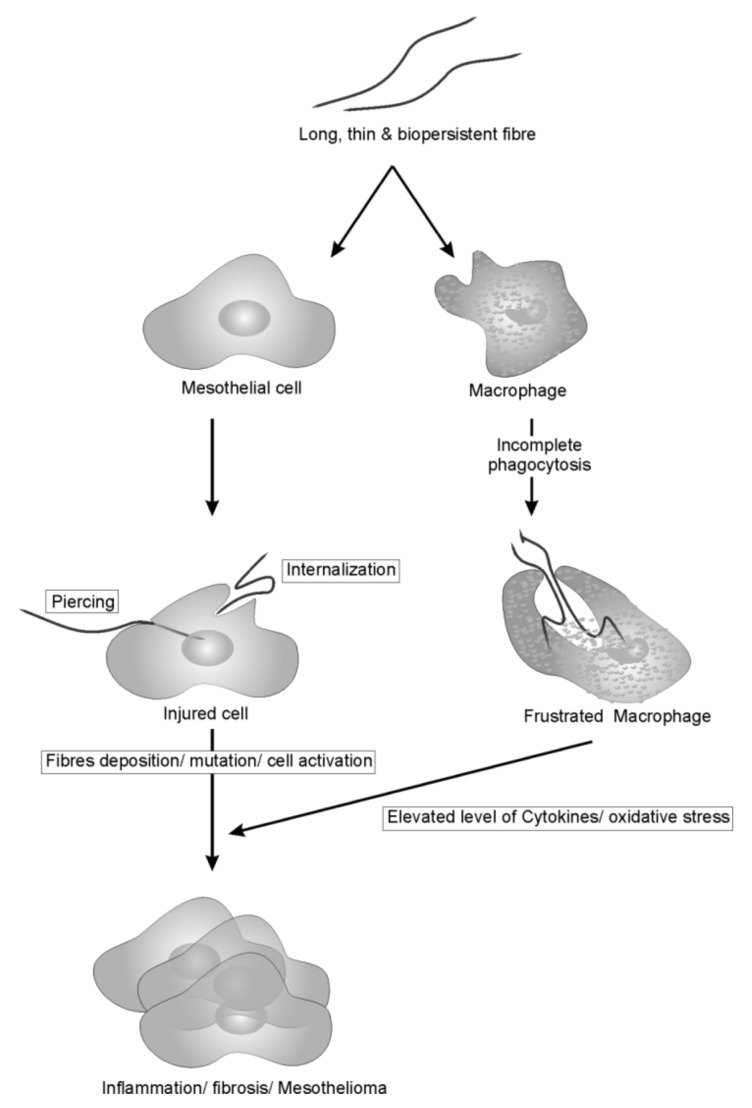
Pathogenicity paradigm of long, thin, bio-persistent fibers on mesothelial cells and macrophage. Fibers, once inhaled, induces cell injury either by piercing or internalization in mesothelial cells, resulting in mutation and cell activation. On the other hands, foreign fibers recognized by macrophages resulted in the incomplete phagocytosis due to length and bio-persistent nature. This incomplete phagocytosis frustrates macrophages that result in elevated level of cytokines and ROS which indirectly associated with the activation of cancer signaling pathways.

**Figure 2 nanomaterials-12-01708-f002:**
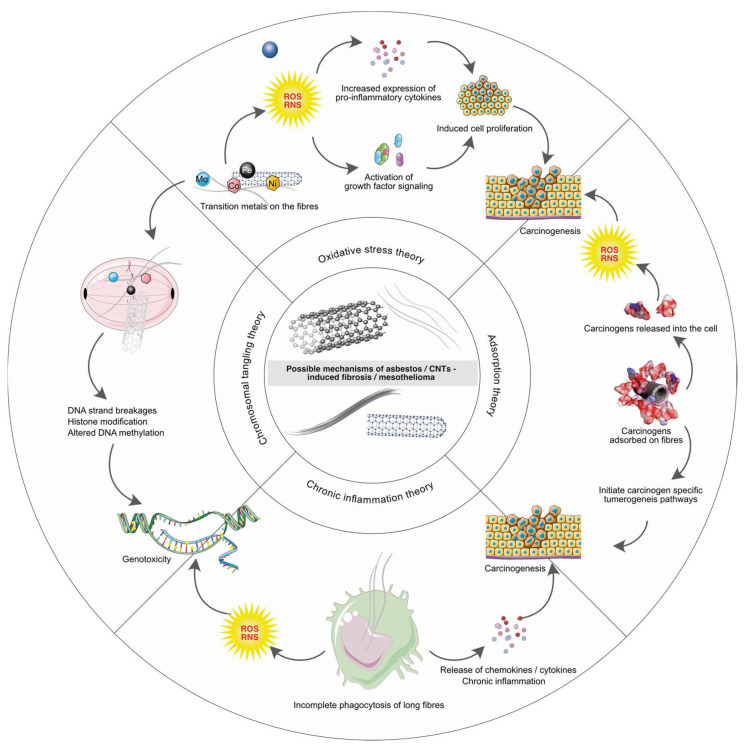
The four possible mechanisms of asbestos/CNTs induced pathogenicity i.e., oxidative stress theory, chromosomal tangling theory, adsorption theory and chronic inflammation theory are highlighted.

**Table 1 nanomaterials-12-01708-t001:** Selected studies related to the asbestos induced genotoxicity.

Fiber Type	Size	Test System	Key Findings	Refs.
Crocidolite asbestos	Length > 5 µm and diameter < 2 µm	8-week-old transgenic F344 rats bearing multiple copies of λlacI shuttle vectors. Mutation frequencies after the administration of 2 and 5 mg of crocidolite were analyzed in DNA of omentum, a relevant target tissue for mesothelioma carcinogenesis.	Reactive oxygen or nitrogen species in crocidolite asbestos fibers induces mutagenesis.	[104]
Crocidolite asbestos	Mean length 10 µm and mean diameter 0.21 µm	Human bronchial epithelial cancer (A549) cell line was exposed to asbestos, silica and TiO_2_ particles to analyze ROS, apoptosis and DNA double-strand breaks.	Crocidolite has a greater carcinogenic potential than silica and TiO_2_, judged by its ability to cause sustained genomic instability in normal lung cells.	[100]
Crocidolite fibers	Length 3.2 ± 1.0 µm and diameter 0.22 ± 0.01 µm	Hamster hybrid (A_L_) cells, containinga standard set of CHO-K1 chromosomes and a single copy of humanchromosome 11 exposed to crocidolite fiber for various periods of time.	Extra-nuclear targets play an essential role in the initiation of oxidativedamage in fiber mutagenesis in mammalian cells.	[105]
Crocidolite fibers	Length > 5 µm and diameter < 3 µm	Transgenic male *LacI* mice to study the mutagenesis potential of asbestos crocidolite. Mice were exposed to an aerosol containing 5.75 mg/m^3^ crocidolite dust for 6 hr/day and 5 consecutive days.	Significant increase of the mutant frequency of lung DNA after nose-only fiber inhalation.	[106]
Chrysotile and Crocidolite asbestos	Length > 4 µm and diameter < 2 µm	Immortalized human SAE cells were treated with chrysotile or crocidolite at concentrations of 0.5, 1, 2 and/or 4 µg/cm^2^ for 12, 24 or 48 h.	Asbestos may initiate mitochondria-associated ROS, which mediate asbestos-induced nuclear mutagenic events and inflammatory signaling pathways in exposed cells.	[107]

**Table 2 nanomaterials-12-01708-t002:** Selected genotoxicity studies related to the exposure of carbon nanotubes.

Fiber Type	Size	Test System	Key Findings	Refs.
SWCNT and MWCNT	SWCNT: (D) < 2 nm, (L) 4–15 μm MWCNT: (D) 10–30 nm, (L) 1–2 μm	Urinary mutagenicity study in male Fischer-344 rats by oral administration with a single dose of 50 mg/kg body weight of SWCNT or MWCNT.	No increase in urinary mutagenicity were observed in rat using Ames test. SWCNTs and MWCNTs were deposited in the lung and induced an acute lung and systemic effect, which was more pronounced in the MWCNT exposure.	[123]
SWCNT and MWCNT	SWCNT: (D) 0.8–1.2 nm, (L) 0.1–1 μm MWCNT: (D) ~80 nm, (L) 10–20 μm	C57BL/6 mice were exposed by pharyngeal aspiration to vehicle, ultrafine carbon black, SWCNTs or MWCNTs at a dose of 40 μg per mouse and sacrificed 4 h postexposure.	Gene expression in lung and blood: Upregulation of genes involved in inflammation, oxidative stress, coagulation, tissue remodeling. Increased percentage of polymorphonuclear leucocytes (PMN) in blood and bronchoalveolar lavage (BAL).	[124]
SWCNT	(D) 10–30 nm, (L) 2–3 μm	Male CD-ICR mice were exposed to SWCNTs using single tail vein injection at a dose of 40 μg/mouse, 200 μg/mouse and 1.0 mg/mouse. Accumulation determination and toxicological assays were carried out after 90 days post-exposure.	Inflammation: dose-dependent thickening of the alveolar lining. Particles deposition were observed even after 3 months.	[79]
SWCNT	Mean diameter 1.8 nm, Medium length 4.4 μm	SWCNTs exposure to rat in single instillation (1.0 mg/kg body weight) or repeated intratracheal instillation (0.2 mg/kg body weight) once a week for five weeks.	Inflammatory response (hemorrhage in the alveolus, infiltration of alveolar macrophages and neutrophiles), but no DNA damage, in the lungs in rats. SWCNTs were not genotoxic in the comet assay following intratracheal instillation in rats.	[125]
SWCNT and MWCNT	SWCNT: (D) 1.2–1.5 nm (L) 2–5 µm MWCNT: (D) 10–30 nm (L) 0.5–50 µm	The mouse macrophage cell line RAW 264.7 were treated with different concentrations of CNTs for 24, 48 or 72 h for cytotoxicity, genotoxicity analysis and detection of ROS.	CNTs exposure increase ROS production and are cyto- and genotoxic to mouse macrophage cell line. Due to CNTs exposure necrosis and chromosomal aberrations were detected, although no inflammatory responses were observed.	[126]
MWCNT	(D) 10–15 nm, ~20 μm	Eight-week-old rats were subjected to whole-body exposure to low (0.01 mg/m^3^), middle (0.1 mg/m^3^), high-concentration (1 mg/m^3^) of MWCNT aerosol and clean air control for 6 h/day for 5 days. Lung cells were analyzed using comet assay for DNA damages on day 0 and 1 month after the exposure.	MWCNTs caused a statistically significant increase in lung DNA damage and genotoxicity at high concentration when compared with the negative control group on day 0 with the lung burden retained for 1 month post exposure.	[127]

## Data Availability

Not applicable.

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
