# Peer review of "Do Carbon Nanotubes and Asbestos Fibers Exhibit Common Toxicity Mechanisms?"

_nanomaterials, 2022, doi:10.3390/nano12101708_

Round 1
Reviewer 1 Report
see the attached file

Author Response
Dear Reviewer, Thank you very much for reviewing our manuscript. We checked the manuscript and corrected all typos.
Reviewer 2 Report
The article makes a bibliographical review of the subject in the works carried out in the last 20 years and alerts to a problem, unfortunately observed several times in the history of science, the industrial adoption of a technology without fully knowing its implications for nature and, in particularly for human health. In this way the work is well written and draws attention to a problem that, at the very least, deserves further investigation.
The manuscript has some typographical errors that should be corrected (line 165,169, table 1, line 248, table 2, line 363).
Citations in the text are marked with () and should be replaced by [] (marked in the pdf document. References are badly formatted according to the journal's rules and some may not be correct {175] , or in the format recommended by the editor, for example [188].

Author Response
Reviewer 2:
The article makes a bibliographical review of the subject in the works carried out in the last 20 years and alerts to a problem, unfortunately observed several times in the history of science, the industrial adoption of a technology without fully knowing its implications for nature and, in particularly for human health. In this way the work is well written and draws attention to a problem that, at the very least, deserves further investigation.
The manuscript has some typographical errors that should be corrected (line 165,169, table 1, line 248, table 2, line 363).
Response: Thank you for pointing out the typographical errors. In the revised version, we have carefully cross-checked the manuscript and corrected all the typographical errors.
Citations in the text are marked with () and should be replaced by [] (marked in the pdf document. References are badly formatted according to the journal's rules and some may not be correct {175] , or in the format recommended by the editor, for example [188].
Response: Thank you for pointing out the issues with reference formatting. In the revised version, we used Zotero software with the reference style of Nanomaterials journal. We also observed few duplications in the review list which is now corrected in the revised version.
Revised manuscript is attached.
